# SEEDFT: STRUCTURE-PRESERVING FUSION FOR MULTI-SEED LLM FINE-TUNING

## ABSTRACT

Fine-tuning large language models exhibits high variance across random seeds, often requiring multiple runs to find the best checkpoint. While ensemble methods can leverage this diversity, they incur prohibitive computational costs during inference, and existing model merging techniques rely on element-wise operations that treat weight matrices as vectors, destroying the geometric structure essential for effective knowledge consolidation. We address this limitation through SeedFT, a training-free fusion method that preserves matrix geometry while consolidating complementary capabilities from multiple seed-specific fine-tuned models. Our approach builds on two key observations: layer-wise fine-tuning updates contain substantial redundancy, with the top 50% of singular directions preserving over 99% of model performance, and different random seeds learn complementary sub-skills within the same task domain. SeedFT operates through structure-preserving aggregation in two stages: first aligning seed-specific updates in a shared SVD-derived subspace, then fusing these aligned representations via orthogonality-constrained optimization with a closed-form solution. Across mathematical reasoning, commonsense reasoning, and code generation benchmarks, SeedFT consistently matches or exceeds the best individual seed while outperforming element-wise baselines.

## 1 INTRODUCTION

Fine-tuning large language models (LLMs) is computationally expensive and exhibits high variance across random seeds (Biderman et al., 2024). Even with identical data and hyperparameters, different initialization and minibatch ordering can lead to substantially different results, often requiring multiple training runs to identify the best checkpoint. While ensembles of these models reduce variance and improve performance (Izmailov et al., 2018), they proportionally increase inference costs—a critical limitation for deployment scenarios with strict latency or resource constraints.

Post-hoc weight-space merging offers an appealing alternative: combining multiple trained checkpoints into a single model that maintains the inference cost of an individual model (Wortsman et al., 2022). However, existing approaches face two fundamental challenges. First, most merging methods apply element-wise operations (averaging, masking, or magnitude-based selection) (Yadav et al., 2023; Yu et al., 2024; Ilharco et al., 2022) that treat weight matrices as flat vectors, ignoring their inherent geometric structure. This simplification can cause destructive interference between parameters, leading to suboptimal fusion (Ainsworth et al., 2022). Second, while most research focuses on multi-task merging—combining models trained on different datasets (Stoica et al., 2024; Wang et al., 2024)—the single-task setting, where models are fine-tuned on the same data with different random seeds, remains underexplored despite offering better initial alignment and unique opportunities for effective fusion.

We investigate this single-task, multi-seed setting and uncover two key insights that motivate our approach. First, we demonstrate that layer-wise fine-tuning updates exhibit significant redundancy: truncating each update to its top 50% singular directions preserves over 99% of the original model's accuracy (Section 3.1). This reveals substantial low-utility capacity in the tail directions that could potentially be repurposed. Second, we show that models fine-tuned with different random seeds develop complementary capabilities, excelling at different sub-tasks within the same domain (Sec-

tion 3.2). For instance, when fine-tuning on mathematical reasoning, different seeds achieve best performance on distinct problem types, with no single seed dominating across all categories.

These complementary findings suggest a natural fusion strategy: leverage the spare capacity in each model's tail directions to incorporate complementary knowledge from other seeds. The challenge lies in combining these models while preserving their geometric structure and avoiding the reintroduction of redundant information. Element-wise averaging would destroy the orthogonal structure learned during training, while simple concatenation would ignore the relationships between different seeds' representations.

We propose *SeedFT*, a training-free fusion method that addresses these challenges through structure-preserving aggregation in weight space. Our approach operates in two stages. First, we align all seed-specific updates in a shared subspace via SVD, expressing each model in common coordinates. Second, we aggregate these aligned representations using an orthogonality-constrained optimization that yields a closed-form solution, ensuring the merged model maintains meaningful geometric structure without redundancy. Unlike element-wise, SeedFT respects the matrix geometry of neural network weights, leading to more effective knowledge consolidation.

We evaluate *SeedFT* across mathematical reasoning, commonsense reasoning, and code generation tasks. Our method consistently outperforms element-wise baselines such as Model Soup (Wortsman et al., 2022), TIES (Yadav et al., 2023), and DARE (Yu et al., 2024) while matching or exceeding the best individual seed. For instance, when fine-tuning LLaMA-2-7B (Touvron et al., 2023) on MetaMathQA (Yu et al., 2023), *SeedFT* achieves 71.0% on GSM8K (Cobbe et al., 2021) and 22.1% on MATH (Hendrycks et al., 2021), compared to 67.2% and 20.4% for the best seed—relative improvements of 5.7% and 8.3% respectively. These gains are achieved without additional training or data, demonstrating the effectiveness of structure-preserving fusion over element-wise approaches.

The main contributions of this paper are:

- We show that leading singular directions of layer-wise updates retain most accuracy, and that different seeds learn complementary sub-skills on the same task.

- We propose SeedFT, a two-stage SVD+Procrustes fusion that aligns seeds in a shared subspace and aggregates their coordinates with a closed-form solution, preserving matrix geometry and adding no inference-time cost.

- SeedFT consistently matches or improves over the best seed and outperforms element-wise merging baselines across math, commonsense, and code benchmarks.

## 2 PRELIMINARIES

### 2.1 MODEL MERGE

Model merging is a technique that combines knowledge from multiple trained models into one unified model. What makes this approach different from ensemble methods is that instead of running several models simultaneously, we compress their combined expertise into a single set of parameters. This leads to better efficiency and often improves how well the model generalizes to new situations. Research in this area has mainly split into two directions.

The first is multi-task merging, where we take models that were each trained for different tasks and combine them into one model that can handle all those tasks, such as TIES (Yu et al., 2024), DARE (Yu et al., 2024), Task Arithmetic (Ilharco et al., 2022), Knots (Stoica et al., 2024), TALL-Masks (Wang et al., 2024), PCB-Merging (Du et al., 2024), CABS (Qi et al.).

The second direction, which hasn't gotten as much attention, is single-task or same-task merging. Here, we merge models that were trained on the same task or very similar ones. The goal is to make the combined model more robust and better at generalizing. Studies have shown this can help models perform better when they encounter data that's a bit different from what they were trained on. Most of the work so far has been in computer vision, such as model soup (Wortsman et al., 2022), SWA (Izmailov et al., 2018), Git Re-basin (Ainsworth et al., 2022), which means there's a lot of room to explore these techniques in other areas like natural language processing.

## 2.2 Singular Value Decomposition (SVD)

For any matrix $A \in \mathbb{R}^{m \times n}$, the singular value decomposition (SVD) writes

$$W = U \Sigma V^\top, \tag{1}$$

where $U \in \mathbb{R}^{m \times m}$ and $V \in \mathbb{R}^{n \times n}$ are orthogonal, and $\Sigma = \text{diag}(\sigma_1, \ldots, \sigma_r, 0, \ldots)$ has nonnegative entries with $\sigma_1 \geq \cdots \geq \sigma_r > 0$. The vectors in $U$ and $V$ form orthonormal directions, and each $\sigma_i$ shows how much $W$ scales the corresponding direction. The rank of $W$ equals the number of nonzero singular values, and the energy satisfies $\|W\|_F^2 = \sum_i \sigma_i^2$.

## 3 Our Method

In this section, we will investigate model fusion in the weight space for a single task across multiple random seeds. The approach is motivated by two observations: first, most performance is retained after low-rank truncation of fully fine-tuned deltas; second, different seeds specialize in complementary sub-skills on the same task. Based on the above analysis, we propose SeedFT, which aims to merge multiple models with different random seeds, and therefore comprises (1) SVD-based subspace alignment on seed delta model weights and (2) an orthogonality-constrained aggregation that fuses the aligned coordinates into one model.

### 3.1 Redundancy in LLM Fine-Tuning

Related work reports that LLM fine-tuning updates are sparse and redundant. For example, retaining roughly half of the parameters with appropriate scaling can preserve much of the original performance (Yu et al., 2024). Against this backdrop, we ask: *to what extent can the performance of a fully fine-tuned model be recovered when each layer-wise update is truncated to its leading singular directions?* A positive answer would indicate sizeable low-utility capacity that could be repurposed in later fusion.

Motivated by this, we truncate each layer-wise $\Delta\theta$ to its leading singular directions and evaluate the resulting model. Concretely, let $\theta_{\text{pre}}$ be the base model and $\theta = \theta_{\text{pre}} + \Delta\theta$ the fully fine-tuned model on dataset $\mathcal{D}$, where $\Delta\theta$ denotes the *layer-wise* parameter deltas. For each layer's delta $\Delta\theta \in \mathbb{R}^{m \times n}$, we compute the SVD $\Delta\theta = U\Sigma V^\top$ with singular values $\{\sigma_k\}_{k \geq 1}$. Form truncated reconstructions by fixing a target rank $r$

$$\widehat{\Delta\theta}(r) \;=\; U_{[:,1:r]} \, \Sigma_{1:r,1:r} \, V_{[:,1:r]}^\top,$$

Then reconstructed model $\tilde{\theta}(r) \;=\; \theta_{\text{pre}} + \widehat{\Delta\theta}(r)$ is evaluated under the same decoding protocol as the full model.

Table 1 demonstrates a rapid accuracy increase at low ranks followed by early saturation: performance improves markedly through 40% rank and effectively plateaus by 50%, at which point the truncated model recovers $\geq 99\%$ of the full-model accuracy on both tasks (66.5 vs. 67.0 on GSM8K; 20.1 vs. 20.3 on MATH), and the average differs by only 0.4 points from the 100% model (43.3 vs. 43.7). Beyond 60–70% rank, incremental gains are negligible, indicating pronounced diminishing returns: informative content is concentrated in the leading spectrum, whereas the residual tail contributes little and thus constitutes capacity that can be reallocated for fusion.

| Model | Task | 10% | 20% | 30% | 40% | 50% | 60% | 70% | 80% | 90% | 100% |
|---|---|---|---|---|---|---|---|---|---|---|---|
| **LLaMA2-7B** | **GSM8K** | 65.0 | 65.0 | 65.4 | 66.3 | 66.5 | 67.2 | 67.5 | 67.0 | 67.1 | 67.0 |
| **LLaMA2-7B** | **MATH** | 14.9 | 18.6 | 19.7 | 19.8 | 20.1 | 20.3 | 20.3 | 20.3 | 20.4 | 20.3 |
| **Average** | | 40.0 | 41.8 | 42.6 | 43.1 | 43.3 | 43.8 | 43.9 | 43.7 | 43.8 | 43.7 |

Table 1: Accuracy of Fine-Tuning LLaMA2-7B on MetaMathQA.

**Implication for Fusion.** The presence of low-utility tail directions indicates repurposable capacity. In Sec. 3.4, we exploit this capacity by aligning a shared subspace across seeds and injecting complementary directions from other seeds, thereby replacing part of the tail with informative structure.

## 3.2 Diverse Ability among Trained Models with Different Random Seeds

Building on Sec. 3.1, which indicates repurposable tail capacity in layer-wise updates, we hypothesize that stochastic variability from different random seeds can supply complementary information. Even under identical data and hyperparameters, optimization noise (e.g., minibatch ordering and initialization) can steer training toward distinct yet aligned representation subspaces. This provides a principled rationale to examine whether seed-wise fine-tuning induces complementary competencies within a single task domain.

Concretely, let $\theta_{\mathrm{pre}}$ be the base model and, for seeds $\{s_i\}_{i=1}^N$, define $\theta_i = \theta_{\mathrm{pre}} + \Delta\theta_i$ via full-parameter fine-tuning on the same dataset $\mathcal{D}$ with identical hyperparameters and schedule. We evaluate under a fixed decoding protocol on a suite of sub-tasks $\mathcal{T} = \{t_1, \ldots, t_M\}$ from the same domain, computing per–sub-task accuracy $\mathrm{Acc}(\theta_i; t)$.

Figure 1 supports complementary specialization: no single seed dominates; column-wise best scores are distributed across seeds, and several sub-tasks exhibit non-negligible $\mathrm{Disp}(t)$. For instance, with LLaMA3-8B fine-tuned on a commonsense reasoning task using three seeds $\{11, 87, 100\}$, one seed leads on *HellaSwag* while another leads on *ARC-C/WinoGrande*, indicating partially complementary, task-relevant directions.

These observations suggest that seed-wise updates $\{\Delta\theta_i\}$ encode partially complementary task-relevant directions. Sec. 3.4 leverages this by aligning a shared subspace and aggregating seed coordinates in a structure-preserving manner so the merged model can inherit strengths from multiple seeds.

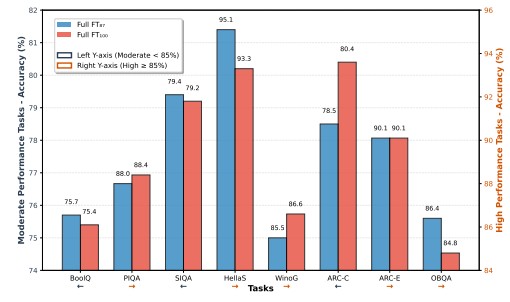

## 3.3 Problem Definition

Sections 3.1 and 3.2 respectively show that layer-wise fine-tuning updates contain low-utility tail directions and that different random seeds acquire complementary sub-skills on the same task. These observations motivate reusing the tail capacity by fusing complementary information across seeds within a single-task setting.

Figure 1: Performance Comparison of LLaMA3-8B Models.

Formally, let $\theta_{\mathrm{pre}}$ denote the base model. For seeds $\{s_i\}_{i=1}^N$, we obtain fully fine-tuned models $\theta_i = \theta_{\mathrm{pre}} + \Delta\theta_i$ on the same dataset with identical hyperparameters and schedule. All operations act per layer on the deltas $\{\Delta\theta_i\}$. The objective is to construct a merged delta

$$\Delta\theta_M = \mathrm{Merge}(\Delta\theta_1, \ldots, \Delta\theta_N), \qquad \theta_M = \theta_{\mathrm{pre}} + \alpha\,\Delta\theta_M, \tag{2}$$

where $\alpha$ is a global scale. This problem differs from traditional multi-task merging at the task level: most prior work merges models trained on different datasets, where alignment is weaker. However, single-task seed models are more strongly aligned and exhibit greater redundancy. It also differs in the fusion space: rather than applying element-wise rules that treat weight matrices as vectors and ignore matrix geometry, we fuse in a shared subspace and preserve orthogonality of coordinates to avoid reintroducing redundancy.

## 3.4 SVD-based Multi-Seed Model Fusion

Motivated by the single-task, multi-seed setting and the redundancy/complementarity observations in Secs. 3.1–3.2, **SeedFT** fuses weight matrix per layer in two stages. **Stage 1 (alignment)** places

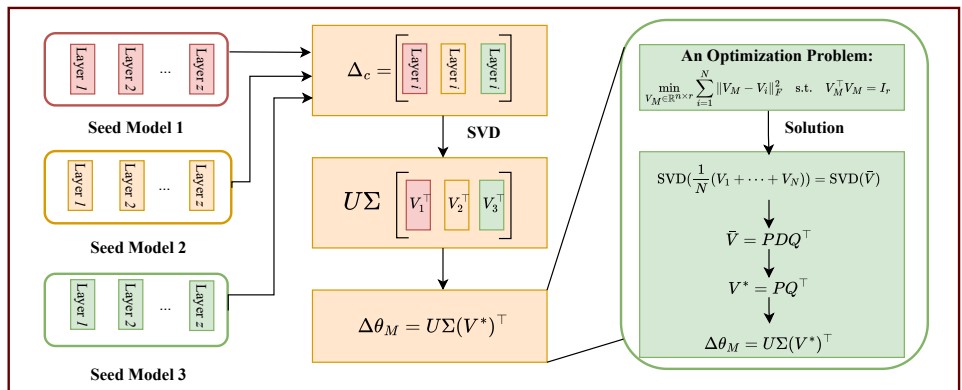

Figure 2: The Pipeline of SeedFT for Full-Parameter Fine-Tuning.

all seed models in a shared basis so that their updates are expressed in the same coordinates. **Stage 2 (structure-preserving aggregation)** merges these aligned coordinates into a single orthonormal coordinate that is close to all seeds. Rather than applying element-wise rules that treat weight matrices as vectors and ignore matrix geometry, we fuse in a shared subspace and preserve orthogonality of coordinates to avoid reintroducing redundancy. We then reconstruct the merged update in the shared basis and assemble the final model.

**Stage 1: Subspace alignment via SVD.** Following the setting in (Stoica et al., 2024), this stage exposes the dominant cross-seed subspace and represents each seed in the same coordinates. For each layer, we concatenate the seed-specific updates along columns to form

$$\Delta_c = \begin{bmatrix} \Delta\theta_1 \,|\, \Delta\theta_2 \,|\, \cdots \,|\, \Delta\theta_N \end{bmatrix} \in \mathbb{R}^{m \times (Nn)},$$

where $N$ is the number of seed models and $\Delta\theta_i \in \mathbb{R}^{m \times n}$. Then, we compute the SVD on $\Delta_c$:

$$\Delta_c = U\Sigma V^\top, \quad U \in \mathbb{R}^{m \times r_*}, \ \Sigma \in \mathbb{R}^{r_* \times r_*}, \ V \in \mathbb{R}^{(Nn) \times r_*},$$

where $r_* = \mathrm{rank}(\Delta_c) \leq \min(m, Nn)$. Partition $V^\top$ into $N$ contiguous blocks of width $n$ (one per seed):

$$V^\top = \begin{bmatrix} V_1^\top \,|\, V_2^\top \,|\, \cdots \,|\, V_N^\top \end{bmatrix}, \qquad V_i \in \mathbb{R}^{n \times r_*}.$$

This defines a shared basis $U\Sigma$ and seed coordinates $\{V_i\}$ such that, for each seed $i$,

$$\Delta\theta_i = U\Sigma V_i^\top$$

In the following section, we will introduce how to fuse in a shared subspace and preserve orthogonality of coordinates to avoid reintroducing redundancy, rather than applying element-wise rules that treat weight matrices as vectors.

**Stage 2: Orthogonal aggregation (structure-preserving fusion).** From Step 1, we write $\Delta\theta_i = U\Sigma V_i^\top$ with $V_i \in \mathbb{R}^{n \times r_*}$. The differences across seeds lie in the right-side coordinates $\{V_i\}$ under the shared basis $U\Sigma$. We want a single merged coordinate $V_M$ that (i) has orthonormal columns (to keep the learned directions and avoid redundancy) and (ii) is, in least squares, close to all seeds. We therefore define an optimization problem to solve:

$$\min_{V_M \in \mathbb{R}^{n \times r}} \sum_{i=1}^{N} \|V_M - V_i\|_F^2 \quad \text{s.t.} \quad V_M^\top V_M = I_r, \qquad r = \min\{\mathrm{rank}(\Delta_c), n\}. \tag{3}$$

Using the Frobenius norm gives a *closed-form* solution. Expanding the sum,

$$\sum_{i=1}^{N} \|V_M - V_i\|_F^2 = N\|V_M\|_F^2 - 2\,\mathrm{tr}(V_M^\top \bar{V}) + \text{const}, \qquad \bar{V} = \tfrac{1}{N}\sum_{i=1}^{N} V_i, \tag{4}$$

and since $V_M^\top V_M = I_r$ implies $\|V_M\|_F^2 = r$, the problem is equivalent to

$$\max_{V_M^\top V_M = I_r} \ \text{tr}(V_M^\top \bar{V}). \tag{5}$$

Let the SVD of $\bar{V}$ be

$$\bar{V} = PDQ^\top, \qquad V^* = PQ^\top, \tag{6}$$

then $V^*$ is the optimizer (orthogonal Procrustes) and serves as the fused coordinate. Intuitively, if each $V_i = V_\star +$ zero-mean noise, $V^*$ is the least-squares (and maximum-likelihood) estimator: it projects the simple average $\bar{V}$ back to the set of matrices with orthonormal columns.

Finally, we form the merged update and the model as

$$\Delta\theta_M = U\Sigma(V^*)^\top, \qquad \theta_M = \theta_{\text{pre}} + \alpha\,\Delta\theta_M. \tag{7}$$

The implementation details are given in Algorithm 1.

## 4 EXPERIMENTS

### 4.1 EXPERIMENTAL SETUP

**Tasks and Datasets.** For mathematical reasoning, we fine-tune on MetaMathQA (Yu et al., 2023) and evaluate on GSM8K (Cobbe et al., 2021) and MATH (Hendrycks et al., 2021). For commonsense reasoning, we use Commonsense-170k for training and evaluate on eight benchmarks: BoolQ (Clark et al., 2019), PIQA (Bisk et al., 2020), SIQA (Sap et al., 2019), HellaSwag (Zellers et al., 2019), WinoGrande (Sakaguchi et al., 2021), ARC-C/E (Clark et al., 2018), and OBQA (Mihaylov et al., 2018). For code generation, we train on Code-Feedback (Zheng et al., 2024) and evaluate on HumanEval (Chen et al., 2021) and MBPP from EvalPlus (Liu et al., 2024).

**Models and Training.** We use LLaMA-2-7B (Touvron et al., 2023), LLaMA-3-8B, Mistral-7B (Jiang et al., 2023), and Gemma-2-9B as base models. For each task, we perform full-parameter fine-tuning with three different random seeds (42, 87, 100) using identical hyperparameters and schedules. Training uses BFloat16 precision with cosine learning rate scheduling and 3% linear warmup over 3 epochs.

**Baselines.** We compare against element-wise fusion methods: Model Soup (Wortsman et al., 2022) (uniform averaging), TIES (Yadav et al., 2023) (interference resolution), and DARE (Yu et al., 2024) (parameter dropping with rescaling). We also report individual seed performance to establish upper bounds.

### 4.2 MATHEMATICAL REASONING

Table 2 presents results on mathematical reasoning tasks. SeedFT consistently outperforms all baselines across model architectures. On LLaMA-2-7B, SeedFT achieves 71.0% on GSM8K and 22.1% on MATH, representing 2.8% and 1.7% improvements over the best individual seed . This improvement is particularly notable given that mathematical reasoning requires precise logical steps where interference between conflicting solutions could be detrimental.

| Model | Task | seed=42 | seed=87 | seed=100 | Model Soup | TIES | DARE | SeedFT |
|---|---|---|---|---|---|---|---|---|
| **LLaMA2-7B** | **GSM8K** | 66.9 | 67.2 | 66.6 | 59.7 | 68.1 | 68.3 | **71.0** (↑ 3.8) |
| **LLaMA2-7B** | **MATH** | 19.0 | 19.4 | 20.4 | 16.9 | 19.8 | 20.1 | **22.1** (↑ 1.7) |
| **LLaMA3-8B** | **GSM8K** | 80.6 | 80.3 | 81.3 | 81.8 | 81.3 | 81.4 | **82.1** (↑ 1.5) |
| **LLaMA3-8B** | **MATH** | 31.8 | 31.5 | 31.6 | 32.8 | 32.0 | 32.1 | **33.3** (↑ 1.4) |
| **Gemma2-9B** | **GSM8K** | 81.0 | 82.2 | 81.5 | 82.9 | 81.9 | 82.2 | **85.0** (↑ 2.8) |
| **Gemma2-9B** | **MATH** | 36.4 | 36.1 | 37.3 | 39.9 | 38.1 | 38.3 | **41.3** (↑ 4.0) |

Table 2: Accuracy of Fine-Tuning LLaMA2-7b, LLaMA3-8B and Gemma2-9B on MetaMathQA.

The results reveal interesting patterns in baseline performance. Model Soup shows inconsistent behavior—improving performance on larger models (LLaMA-3-8B, Gemma-2-9B) but degrading significantly on LLaMA-2-7B (59.7% vs. 67.2% best seed on GSM8K). This instability likely stems from destructive interference when averaging parameters without considering their geometric relationships. TIES and DARE show more stable performance but achieve smaller gains, suggesting that element-wise interference resolution is insufficient to fully exploit complementary knowledge.

SeedFT's superior performance demonstrates that preserving matrix structure during fusion enables more effective knowledge consolidation. The method successfully combines different seeds' complementary problem-solving strategies while avoiding the geometric distortions that plague element-wise approaches.

### 4.3 COMMONSENSE REASONING

Table 3 presents results for commonsense reasoning on LLaMA2-7B and LLaMA3-8B, where the subscript numbers indicate different random seeds used during fine-tuning. SeedFT achieves improved average accuracy across eight benchmarks, with notable gains on WinoGrande (87.7% vs. 86.6% best seed) and ARC-C (82.2% vs. 80.4% best seed).

The commonsense domain particularly benefits from SeedFT's orthogonalization approach because fine-tuning with different seeds often produces models with complementary strengths across reasoning types. Our analysis reveals that no single seed dominates all subtasks—some models excel at physical reasoning while others perform better on social inference tasks. This observation aligns with our earlier findings that orthogonalization helps balance different reasoning capabilities by preventing the dominance of particular update directions, allowing SeedFT to consolidate these diverse capabilities into improved performance.

| Model | Method | BoolQ | PIQA | SIQA | HellaS | WinoG | ARC-C | ARC-E | OBQA | Average |
|---|---|---|---|---|---|---|---|---|---|---|
| **LLaMA3-8B** | **Full FT**$_{42}$ | **75.7** | **89.1** | 79.8 | 94.5 | 86.5 | 80.1 | 91.0 | 88.2 | 85.6 |
| **LLaMA3-8B** | **Full FT**$_{87}$ | **75.7** | 88.0 | 79.4 | 95.1 | 85.5 | 78.5 | 90.1 | 86.4 | 84.8 |
| **LLaMA3-8B** | **Full FT**$_{100}$ | 75.4 | 88.4 | 79.2 | 93.3 | 86.6 | 80.4 | 90.1 | 84.8 | 84.8 |
| **LLaMA3-8b** | **Model Soup** | 75.3 | **89.1** | 81.0 | 96.0 | 84.1 | 81.9 | 91.9 | 88.4 | 86.0 |
| **LLaMA3-8b** | **TIES** | 75.3 | 89.0 | 79.8 | 95.9 | 88.0 | 81.6 | 91.8 | 86.4 | 86.0 |
| **LLaMA3-8B** | **DARE** | 75.6 | 89.0 | 79.8 | 95.1 | 86.6 | 81.6 | 91.0 | **88.8** | 85.9 |
| **LLaMA3-8B** | **SeedFT** | 75.6 | **89.1** | **81.1** | **96.1** | **87.7** | **82.2** | **92.1** | **88.8** | **86.6** ($\uparrow$ 1.0) |

Table 3: Accuracy of Fine-Tuning LLaMA3-8B on Commonsense-170k.

### 4.4 CODE GENERATION

Table 4 and Table 10 demonstrate SeedFT's effectiveness on structured code generation tasks requiring precise syntax and logic. The method consistently outperforms individual seeds and element-wise baselines across both model architectures.

On LLaMA-3-8B, SeedFT achieves 58.8% average accuracy, compared to 57.6% for the best individual seed and 57.7% for Model Soup. The improvements are particularly evident on enhanced benchmarks: HumanEval+ increases from 52.4% to 54.3%, representing a 1.9% gain, while MBPP+ improves from 56.1% to 56.6%. These enhanced benchmarks contain more rigorous test cases, indicating that SeedFT preserves the precise parameter relationships necessary for correct code generation.

The benefits are more substantial on LLaMA-2-7B, with SeedFT achieving 36.7% compared to 35.0% for the best individual seed. HumanEval performance increases from 34.8% to 38.4%, representing a 10.3% relative improvement. This suggests that structure-preserving fusion is particularly beneficial for consolidating complementary coding strategies in models with limited representational capacity.

| Model | Method | HumanEval | HumanEval+ | MBPP | MBPP+ | Average |
|-------|--------|-----------|------------|------|-------|---------|
| **LLaMA2-7B** | **Full FT**$_{42}$ | 31.7 | 29.9 | 41.5 | 33.3 | 34.1 |
| **LLaMA2-7B** | **Model Soup** | 31.7 | 30.5 | 41.5 | 34.1 | 34.5 |
| **LLaMA2-7B** | **TIES** | 36.6 | 33.5 | 39.2 | 32.8 | 35.5 |
| **LLaMA2-7B** | **DARE** | 34.1 | 32.3 | 40.5 | 33.3 | 35.1 |
| **LLaMA2-7B** | **SeedFT** | **38.4** | **35.4** | **39.4** | **33.6** | **36.7** ($\uparrow$ 2.6) |
| **LLaMA3-8B** | **Full FT**$_{42}$ | 53.0 | 49.4 | 64.8 | 55.3 | 55.6 |
| **LLaMA3-8b** | **Model Soup** | 56.7 | 53.0 | 65.1 | 56.1 | 57.7 |
| **LLaMA3-8b** | **TIES** | 56.1 | 51.8 | 65.9 | 56.9 | 57.7 |
| **LLaMA3-8B** | **DARE** | 57.9 | 53.0 | 65.3 | 56.3 | 58.1 |
| **LLaMA3-8B** | **SeedFT** | **57.9** | **54.3** | **66.4** | **56.6** | **58.8** ($\uparrow$ 3.2) |

Table 4: Accuracy of Fine-Tuning LLaMA2-7b and LLaMA3-8B on CodeFeedback.

## 4.5 TRAINING EFFICIENCY ANALYSIS

A natural concern with SeedFT is the computational cost of training multiple seed models. To address this, we compare SeedFT's multi-seed approach against extended single-seed training using equivalent total training time. Specifically, we evaluate SeedFT (3 seeds, 3 epochs each) against single-seed training with extended epochs (6 and 9 epochs) on LLaMA2-7B using MetaMathQA.

Table 5 demonstrates that simply extending training epochs yields minimal improvement and eventually leads to performance degradation. Single-seed training with 6 epochs shows slight decline from 67.2% to 65.6% on GSM8K and from 19.4% to 18.2% on MATH. With 9 epochs, performance drops further to 64.4% and 16.9% respectively, indicating overfitting.

| Model | Task | Epoch = 3 | Epoch = 6 | Epoch = 9 |
|-------|------|-----------|-----------|-----------|
| **LLaMA2-7B** | **GSM8K** | **67.2** | 65.6 | 64.4 |
| **LLaMA2-7B** | **MATH** | **19.4** | 18.2 | 16.9 |

Table 5: The Experiments with More Models for model merge.

In contrast, SeedFT achieves 71.0% on GSM8K and 22.1% on MATH with the same total computational budget (9 epochs total across 3 seeds). This represents a substantial advantage: SeedFT improves performance by 5.8% on GSM8K and 2.7% on MATH compared to the baseline 3-epoch training, while extended single-seed training actually degrades performance. These results demonstrate that the diversity from different random seeds provides complementary knowledge that cannot be obtained through longer training of a single initialization, making SeedFT a computationally efficient strategy for performance improvement.

| Model | Task | Method | Training Loss | Test Accuracy |
|-------|------|--------|---------------|---------------|
| **LLaMA2-7B** | **MetaMathQA** | **Adam** | 0.0603 | 43.3 |
| **LLaMA2-7B** | **MetaMathQA** | **SeedFT** | 0.0655 | 44.7 |

Table 6: The Experiments with Generalization Ability Analysis.

## 4.6 GENERALIZATION ANALYSIS

Weight-space merging methods are often claimed to find flatter minima that improve generalization ability (Wortsman et al., 2022). To investigate whether SeedFT exhibits similar generalization benefits, we analyze the relationship between training loss and test performance, comparing individual seed models with their SeedFT-merged counterparts. Preliminary observations suggest that SeedFT

models can achieve a higher training loss while achieving superior test performance compared to individual seeds. This pattern illustrates that SeedFT can increase the test accuracy through sacrificing training performance, which helps alleviate overfitting and improve generalization.

## 4.7 ABLATION STUDIES

We conduct ablation studies to understand the key design choices in SeedFT, examining both the optimal number of seeds and the contribution of different layer types to the overall performance improvements.

**Effect of Number of Seeds.** To determine the optimal number of seeds for fusion, we evaluate SeedFT with 2-6 seed models on mathematical reasoning using LLaMA2-7B. Table 7 shows that performance initially increases substantially from 1 to 3 seeds (66.6% to 71.0% on GSM8K), peaks at 4 seeds (71.6% GSM8K, 21.4% MATH), then exhibits diminishing returns or slight degradation with additional seeds.

This behavior aligns with our theoretical understanding: early seeds contribute complementary knowledge that fills gaps in the leading singular directions, but subsequent seeds increasingly overlap with existing representations. The peak at 4 seeds suggests an optimal balance between knowledge diversity and redundancy management, where additional seeds begin to reintroduce noise rather than useful complementary information.

| Model | Task | N = 1 | N = 2 | N = 3 | N = 4 | N = 5 | N = 6 |
|---|---|---|---|---|---|---|---|
| **LLaMA2-7B** | **GSM8K** | 66.6 | 70.5 | 71.0 | 71.6 | 71.9 | 71.0 |
| **LLaMA2-7B** | **MATH** | 20.6 | 20.5 | 22.1 | 21.4 | 20.8 | 21.1 |
| **Average** | | 43.6 | 45.5 | **46.6** | 46.5 | 46.4 | 46.1 |

Table 7: Model Merging Performance with Varying Number of Models (N=1 to N=6).

**Layer-wise Contribution Analysis.** To understand which components contribute most to SeedFT's improvements, we analyze the effect of applying fusion to different layer types individually.

Table 8 shows that feed-forward network layers, particularly the Up projection, contribute most significantly to performance gains. The Up projection achieves 69.5% on GSM8K and 20.1% on MATH, outperforming the baseline and all other individual layer types. Gate projection also shows notable improvement (67.7% GSM8K), while attention layers (Q, K, V, O) demonstrate more modest contributions. This finding aligns with recent work indicating that FFN layers primarily store task-specific knowledge, making them optimal targets for cross-seed knowledge consolidation.

| Model | Task | Q | K | V | O | Gate | Up | Down | Full FT |
|---|---|---|---|---|---|---|---|---|---|
| **LLaMA2-7B** | **GSM8K** | 67.2 | 66.6 | 67.2 | 65.8 | 67.7 | **69.5** | 66.6 | 67.2 |
| **LLaMA2-7B** | **MATH** | 19.4 | 19.7 | 19.3 | 19.7 | 19.7 | **20.1** | 19.3 | 19.4 |

Table 8: Layer-wise ablation study of SeedFT on LLaMA2-7B. Each column shows the result of applying SeedFT to specific layer types, with Full FT as the baseline.

## 5 CONCLUSION

In this paper, we propose SeedFT, a structure-preserving fusion method that combines complementary capabilities from multiple seed-specific fine-tuned models. We demonstrate that fine-tuning updates contain substantial redundancy, with different random seeds learning complementary sub-skills within the same task domain. SeedFT leverages this redundancy by aligning seed-specific updates in a shared SVD-derived subspace and fusing them through orthogonality-constrained optimization. Experiments across mathematical reasoning, commonsense reasoning, and code generation show that SeedFT consistently matches or exceeds the best individual seed while outperforming

element-wise baselines, creating stronger unified models without additional training or inference costs.

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

# 6 APPENDIX

## 6.1 USAGE OF LARGE LANGUAGE MODELS

In this work, all research concepts, technical approaches, experimental procedures, and analytical findings originated from the authors' independent research and collaborative discussions. The use of large language models was limited exclusively to text editing and language enhancement purposes. We employed LLMs to improve the readability and linguistic quality of our manuscript, including corrections to grammar, syntax, and phrasing. However, the fundamental research contributions, including problem formulation, solution design, experimental validation, and result interpretation, were developed without any assistance from generative AI tools. The scientific content and intellectual merit of this paper represent purely human-driven research efforts by the author team.

## 6.2 IMPLEMENTATION DETAILS SETTING

Training is conducted on NVIDIA H100 and H200 GPUs using BFloat16 precision. We set the weight decay to 0 and employ a cosine learning rate scheduler with a linear warmup ratio of 0.03. For evaluation, we utilize vLLM (Kwon et al., 2023) to ensure efficient and scalable inference during testing.

| Model | Dataset | LR | LR Scheduler | Warmup | Epochs | Batch Size |
|-------|---------|-----|--------------|--------|--------|------------|
| **LLaMA2-7B** | MetaMathQA | 1e-5 | cosine | 300 | 3 | 128 |
| **LLaMA2-7B** | commonsense-170k | 1e-5 | cosine | 300 | 3 | 32 |
| **LLaMA2-7B** | Code-Feedback | 1e-5 | cosine | 300 | 3 | 32 |
| **LLaMA3-8B** | MetaMathQA | 5e-6 | cosine | 300 | 3 | 128 |
| **LLaMA3-8B** | commonsense-170k | 5e-6 | cosine | 300 | 3 | 32 |
| **LLaMA3-8B** | Code-Feedback | 5e-6 | cosine | 300 | 3 | 32 |

Table 9: Implementation Details of the experiments on MetaMathQA, commonsense-170k and Code-Feedback.

## 6.3 PSEUDOCODE: SEEDFT

---
**Algorithm 1** `SeedFT`
---
**Require:** Base weights $\theta_{\text{pre}}$, seed deltas $\Delta\theta_1, \ldots, \Delta\theta_N$, scaling factor $\alpha$
**Ensure:** Merged model weights $\theta_M$
1: **Concatenate (columns)**: $\Delta_c \leftarrow [\, \Delta\theta_1 \mid \cdots \mid \Delta\theta_N \,]$
2: **Compute SVD**: $\Delta_c = U\Sigma V^\top = U\Sigma [\, V_1 \mid V_2 \mid \ldots \mid V_N \,]^\top$
3: **Average block coordinates**: $\bar{V} \leftarrow \frac{1}{N}\sum_{i=1}^{N} V_i$
4: **Polar step (Procrustes)**: $\bar{V} = PDQ^\top$; $\quad V^* \leftarrow PQ^\top$
5: **Reconstruct merged delta**: $\Delta\theta_M \leftarrow U\Sigma(V^*)^\top$
6: **Assemble merged model**: $\theta_M \leftarrow \theta_{\text{pre}} + \alpha \cdot \Delta\theta_M$
7: **return** $\theta_M$
---

## 6.4 BASELINE METHODS

**Model Soup (Wortsman et al., 2022)**: Model Soup shows that instead of picking one "best" fine-tuned checkpoint, we can average the weights of several runs (all starting from the same pretrained model) and often get higher accuracy and better robustness, with no extra inference cost.

**Task Arithmetic (Ilharco et al., 2022)**: Task vectors are created by subtracting the weights of a pre-trained model from the fine-tuned version. These vectors can be manipulated through simple

arithmetic operations (e.g.negation, addition) to steer model behavior, create multi-task models, or improve performance without additional data.

**TIES (Yu et al., 2024)**: TIES-MERGING reduces interference by addressing redundant parameter values and sign disagreements across models through a three-step process. This method consistently outperforms other merging techniques across various domains.

**DARE (Yu et al., 2024)** : DARE drops a large portion of update parameters and rescales the remaining ones, maintaining performance while reducing redundancy in fine-tuning.

## 6.5 THE EXPERIMENTAL RESULTS ON CODE GENERATION TASK

In this section, we provide the full results of fine-tuning LLaMA2-7B and LLaMA3-8B on code generation task. In table 10, we also provide the results of each seed model.

| Model | Method | HumanEval | HumanEval+ | MBPP | MBPP+ | Average |
|---|---|---|---|---|---|---|
| **LLaMA2-7B** | **Full FT$_{42}$** | 31.7 | 29.9 | 41.5 | 33.3 | 34.1 |
| **LLaMA2-7B** | **Full FT$_{87}$** | 34.8 | 32.3 | 40.2 | 32.8 | 35.0 |
| **LLaMA2-7B** | **Full FT$_{100}$** | 32.3 | 31.1 | 40.7 | 34.1 | 34.6 |
| **LLaMA2-7B** | **Model Soup** | 31.7 | 30.5 | 41.5 | 34.1 | 34.5 |
| **LLaMA2-7B** | **TIES** | 36.6 | 33.5 | 39.2 | 32.8 | 35.5 |
| **LLaMA2-7B** | **DARE** | 34.1 | 32.3 | 40.5 | 33.3 | 35.1 |
| **LLaMA2-7B** | **SeedFT** | 38.4 | 35.4 | 39.4 | 33.6 | 36.7 |
| **LLaMA3-8B** | **Full FT$_{42}$** | 53.0 | 49.4 | 64.8 | 55.3 | 55.6 |
| **LLaMA3-8B** | **Full FT$_{87}$** | 57.3 | 52.4 | 64.6 | 56.1 | 57.6 |
| **LLaMA3-8B** | **Full FT$_{100}$** | 56.7 | 52.4 | 64.3 | 55.8 | 57.3 |
| **LLaMA3-8b** | **Model Soup** | 56.7 | 53.0 | 65.1 | 56.1 | 57.7 |
| **LLaMA3-8b** | **TIES** | 56.1 | 51.8 | 65.9 | 56.9 | 57.7 |
| **LLaMA3-8B** | **DARE** | 57.9 | 53.0 | 65.3 | 56.3 | 58.1 |
| **LLaMA3-8B** | **SeedFT** | 57.9 | 54.3 | 66.4 | 56.6 | 58.8 |

Table 10: Accuracy of Fine-Tuning LLaMA2-7b and LLaMA3-8B on CodeFeedback.

