# OpenReview forum: "SeedFT: Structure-Preserving Fusion for Multi-Seed LLM Fine-Tuning"
_ICLR.cc/2026/Conference — Submitted to ICLR 2026_

### Official Review · Reviewer_5M3C · 2025-10-20

**Soundness:** 2
**Presentation:** 3
**Contribution:** 2
**Rating:** 2
**Confidence:** 4

**Summary:**

The paper studies single-task multi-seed model merging on large language models, proposing SeedFT. Starting from the claim that element-wise merging of layer-wise weight matrices leads to sub-optimal fusion, the authors propose to merge structured weight matrices in a shared subspace. Given a layer, using SVD over structured weight matrices they obtain such shared basis and they derive in closed-form the solution to an orthogonality-constrained optimization problem that avoids redundancy in each merged weight matrix, across seeds. The intuition on their design stems from the fact that (layer-wise) pseudo-gradients/task vectors are typically low-rank and, apparently, fine-tuning the same task may/may not improve performance on held-out tasks depending on the seed. Experiments are carried on four LLMs over standard NLP tasks.

**Strengths:**

- The observations in Sec. 3.2 are interesting, as the same phenomenon is also observed when fine-tuning vision pre-trained models.
- The problem posed in the paper (namely, analyzing the feasibility of supporting also multiple held-out tasks by reusing knowledge of different training runs) is clear and the way the authors tackle this (SeedFT) is solid.
- The paper is well-written for the most part and the figures/plots are good for supporting the explanations. Also, the mathematical notation is useful and properly used.

**Weaknesses:**

Although the paper poses a clear goal to advancing single-task multi-seed model merging, there are significant shortcomings that require attention:

**W1.**

First of all, the central claim regarding that element-wise merging "leads to suboptimal fusion" (LL44-45) is not supported neither theoretically nor numerically throughout the paper (also, the reference to Ainsworth et al., 2022 at L45 refers to a method that purposefully wants to merge different initializations, which is different from your setting).

Furthermore, looking at the results in Section 4, Model Soup, TIES and DARE (which all implement element-wise merging operations) are very competitive: it is unclear which are the benefits of switching to more convoluted (perhaps, more costly in terms of compute?) approaches like SeedFT. This further weakens the central argument of the paper, and I feel additional work is needed to strengthen the core motivation of the paper at this time.

-------------

**W2.**

Regarding novelty, the results of Table 1 (i.e. the low-rank nature of updates) is well-known in literature [1,2] (see eg. Fig. 2 of [1]). Also, SeedFT is very similar to the method in [1], although SeedFT is only applied to single-task multi-seed model merging. I'd advise the authors to clearly highlight the differences and compare SeedFT against theirs in the setting of single-task multi-seed model merging.

Furthermore, the method/setting are seemingly not tied to a specific modality. So, it could be beneficial to analyze also some vision experiments (eg. taking inspiration from any setting in [1]).

-------------

**W3.**

The results of Figure 1 could have been substantiated with further evidence. Although it is clear that some seeds allow for slightly higher held-out task performance (perhaps, due to alignments in the loss landscapes of different tasks?), using just two seeds feels underwhelming. I'd suggest the authors to try multiple seeds.

-------------

**Minor Weaknesses.**

- L35 & L171: I'm not sure why the authors stress the point about "different initialization" twice, when the paper assumes to always carry out fine-tuning starting from the same pre-training every time. I'd suggest to remove this.

- I'd suggest to reframe the contributions, as the second and third read as one single contribution (i.e. your proposed framework).


-------------

**_References:_**

[1] Gargiulo, Antonio Andrea, et al. "Task singular vectors: Reducing task interference in model merging." CVPR 2025.

[2] Hu, Edward J., et al. "Lora: Low-rank adaptation of large language models." ICLR 2022.

**Questions:**

Thanking in advance for their response, I'd kindly invite the authors to address the points raised in the Weaknesses section of this review.

In addition, I'd kindly ask the following question:

- Given the general purpose pre-training, which presumably locates the initial pre-trained model in an already good local minima for a diverse array of downstream tasks, it is unclear whether each fine-tuned checkpoint arrives at a different point in the loss landscape. A visualization of this could be useful/interesting to have.

---

### Official Review · Reviewer_sfFf · 2025-10-24

**Soundness:** 2
**Presentation:** 3
**Contribution:** 2
**Rating:** 2
**Confidence:** 4

**Summary:**

This paper focuses on fusing different fine-tuned weight updates in the single-task, multi-seed setup. It is observed that the model performance is preserved from the top 50% of singular directions of weight update, and that varying random seeds grants the model various knowledge pertaining to the same task. The authors propose SeedFT, which relies on the truncated SVD to pick up and aggregate important information. Experiments are conducted on mathematical reasoning, commonsense reasoning, and code generation tasks to illustrate a consistent performance gain.

**Strengths:**

1. The observation of redundancy and diverse ability in fine-tuned parameters well motivates the SVD-based decomposition and fusion method.
2. The orthogonal aggregation minimizing the distance between $V_M$ and $V_i$ has an analytical solution.
3. Numerical results exhibiting consistent improvement are promising.

**Weaknesses:**

1. Both of the two keys observations presented in this paper have already been reported in prior works in the context of low-rank adaptation (LoRA), yet these highly related studies are neither mentioned nor compared in the current manuscript. Specifically, the finding that performance is largely preserved within the major singular vectors has been observed in [1, 2, 3], while the diversity in model capacity arising from different random seeds has been discussed in [4].
2. Given that 50% singular vectors are redundant, a natural idea would be to adopt LoRA for parameter-efficient adaptation. Indeed, this idea has already been explored in SeedLoRA [4], which also focuses on the single-task, multi-seed setting. However, the current paper claims this to be an underexplored setting. In fact, this paper appears to share several similarities with SeedLoRA, including the core observations and motivations [4, Section 3.1], the problem formulation [4, Section 3.2], and the use of SVD-based low-rank decomposition [4, Section 3.4]. Consequently, the contribution of this paper may be viewed primarily as an extension of SeedLoRA to full fine-tuning.
3. The rationale for using a shared right singular space in Stage 1 remains unclear. An alternative would be to stack the rows of $\Delta \theta_i$ and partition the rows of $U$. More theoretical justification and ablation studies are necessary to clarify the motivation and effectiveness of this design choice.

[1] V. Lialin et al., "Relora: High-rank training through low-rank updates", in ICLR, 2024.
[2] T. Jiang et al., "Mora: High-rank updating for parameter-efficient fine-tuning", arXiv preprint, 2024.
[3] Q. Huang et al., "HiRA: Parameter-efficient hadamard high-rank adaptation for large language models", in ICLR, 2025.
[4] Y. Liu et al., "SeedLoRA: A Fusion Approach to Efficient LLM Fine-Tuning", in ICML, 2025.

**Questions:**

See above.

---

### Official Review · Reviewer_Y1Wd · 2025-10-27

**Soundness:** 2
**Presentation:** 3
**Contribution:** 2
**Rating:** 2
**Confidence:** 3

**Summary:**

This paper studies the model merging problem under the single-task multiple seeds scenario and proposes a fusion method called SeedFT. The core motivation of SeedFT is the redundancy of fine-tuning updates and the complementary sub-skills learned by weights trained using different seeds. SeedFT aims to preserve the matrix geometry after model merging and uses SVD to first align weights in a shared subspace, then to fuse weights using a closed-form solution from the Orthogonal Procrustes Problem. Experiments on reasoning and code generation benchmarks show that SeedFT can lead to weights with better performance.

**Strengths:**

The writing of this paper is easy to follow and of good quality.

The experiments in this paper cover various benchmarks.

The method has clear motivation, and the implementation (if open-sourced) can be valuable and practical for the community.

The performance of models merged using SeedFT is better compared with baseline models and methods.

**Weaknesses:**

One major weakness of this project is that several claims are not well supported by the experimental results.

The basic assumption is that different seeds can learn different abilities for the same task. The work also claims that SeedFT can combine complementary capabilities. However, the results only show that the performance is improved marginally compared with a single model, without analyzing or showing which capabilities are combined and which seed leads to models better at which sub-skills. Based solely on the final performance scores, it is hard to attribute the improved performance to the sub-skill justifications, thus weakening the claims from this paper.

Besides, the related work (line 104) claims that this can be more robust, but the experiments did not show enough proof. There is only a short paragraph (4.6) supporting this claim using loss and performance comparison.

Another major weakness of this paper is the lack of analysis to support the claims above and show the advantage of SeedFT. For instance, this paper can discuss the detailed performance differences between each seed model and how such differences validate the sub-skill claims. Case studies showing which problems are correctly solved by SeedFT yet other methods failed to solve will also help to validate the advantage of SeedFT.

Moreover, one concern is about the randomness and the guarantees of better performance from merging random seed models. If different sets of experiments with different random seed groups also show consistent improved results, the benefits of SeedFT will be better validated.

Other minor points: 1) Line 301 is an incomplete sentence. 2) Figure 1 is too small and hard to read.

**Questions:**

Are there any specific reasons why SeedFT is limited to the single-task multi-seed setting? Does it have limitations when applied to the multi-task setting?

How large is the computation cost of SeedFT? More details on the time and resource efficiency would be valuable to this project.

---

### Official Review · Reviewer_HkHB · 2025-10-28

**Soundness:** 2
**Presentation:** 2
**Contribution:** 2
**Rating:** 2
**Confidence:** 4

**Summary:**

- The paper claims that fine-tuning large models is sensitive to the random seed and fine-tuned parameters using different seeds show better performance on different subsets of data
- It proposes a new method called SeedFT to merge the fine-tuned parameters trained using different seeds into a single model.
- SeedFT merges the task vectors (fine-tuned weights - pretrained weights) obtained using different seeds using an extension of [1].
- As in [1], the different task vectors are concatenated along the rows and decomposed using SVD: $[\Delta W_1; \Delta W_2; \ldots; \Delta W_k] = U \Sigma V^\top$, where $V^\top=[V_1^\top, V_2^\top, \ldots, V_k^\top]$, and $U\Sigma$ provides a shared basis for all the task vectors.
- Different $V_i$ are merged using a closed form solution by framing it as an Orthogonal Procrustes problem.

**Strengths:**

1. The model provides qualitative analysis of how different seeds lead to higher performance on different subsets of data.
2. The paper proposes a new method to merge the task vectors obtained using different seeds into a single model. This is done by extending the Knots merging method [1] to merge the $V_i$ matrices obtained using different seeds.
3. The method is evaluated on multiple datasets (arithmethic reasoning, commonsense reasoning, code generation) and the merged model shows better performance than the original fine-tuned models using a single seed.

**Weaknesses:**

1. Equations 3-7 (analytical solution for merging multiple $V_i$ matrices) are the core contribution of the paper, but the paper does not provide a detailed explanation of how these equations are derived. While the standard solution for the Orthogonal Procrustes problem is well known for single matrix approximation, the paper does not provide the details of how the closed form solution for multiple matrices is derived. This makes the paper hard to parse.
2. The core merging method is framed as SVD of the concatenated task vectors followed by merging the $V_i$ matrices using an Orthogonal Procrustes problem. However, Knots [1] itself proposed a general framework of performing SVD followed by using any arbitrary merging method to merge the $V_i$ matrices. Hence, the contribution of the paper in terms of a new merging method is limited. Without a proper comparison (elaborated in point 3 below), it is hard to assess the quality of the method.
2. The baselines are not comparable

    - SeedFT is an extension of the Knots merging method [1], which itself experiments with multiple merging methods to merge the $V_i$ matrices. However, the paper only compares with element-wise merging methods, not merging methods applied to the $V_i$ matrices. Hence, a comparable baseline will be using the merging methods (as done in the experiments of [1]) and not element-wise merging methods used in the paper.

    - If the proposed method to merge $V_i$ matrices cannot perform better than existing methods, then it casts doubt upon the quality of the method and in effect, the contributions of the paper.
3. The experiments are done with only one set of seeds. For the claim that merging done with SeedFT performs better than original fine-tuned models, the experiments have to be repeated with multiple set of seeds (like merging models trained with 42/43/44, merging models trained with 100/101/102, or any random combination of seeds) to evaluate the stability of the method across any random choice of seeds.
5. The merging method is not tested for merging models for multiple tasks, and hence the applicability of the merging method itself is limited. Without these experiments, the contributions of the paper are limited.

## Minor:
6. The paper can benefit from more proofreading. For e.g. there are multiple references to POME (lines 340, 348, 472, 473), which is never defined in the paper. Line 301 is incomplete.

---
## References
[1] George Stoica, Pratik Ramesh, Boglarka Ecsedi, Leshem Choshen, Judy Hoffman. "Model merging with SVD to tie the Knots", ICLR25

**Questions:**

1. Why do the results on CodeFeedback only report the performance of the original fine-tuned models with only one seed (42)? Shouldn't the results be reported for multiple seeds like that done in the experiments on other datasets?

---

### Meta-Review · Area_Chair_Pze1 · 2025-12-02

**Summary:**

This paper proposes SeedFT, an SVD-based model-merging method designed for the single-task, multi-seed fine-tuning setting. It decomposes seed-specific task vectors into a shared subspace and uses a closed-form Orthogonal Procrustes solution to merge them. Experiments on reasoning and code-generation benchmarks show small improvements over individual models.

Across four reviews, the paper received uniformly negative assessments (all scores of 2). While reviewers found the motivation intuitive and the method straightforward, they identified substantial concerns regarding novelty, missing baselines, unclear mathematical derivations, and insufficient experimental validation of the paper’s core claims. With no rebuttal, there is no reason to expect score changes.

**Reviewer Concerns:**

Unresolved:
* Core mathematical derivation is insufficiently explained.
* Contribution appears incremental and overlaps heavily with Knots and SeedLoRA.
* Baselines are not comparable: missing merging methods that operate in the same SVD subspace.
* Experiments are limited to one set of seeds and do not validate claims about complementary sub-skills or robustness.
* Related work on LoRA-based merging and seed diversity is incomplete.

**Reviewer Scores:**

* Reviewer HkHB (2): unchanged
* Reviewer Y1Wd (2): unchanged
* Reviewer sfFf (2): unchanged
* Reviewer 5M3C (2): unchanged

---

### Decision · Program_Chairs · 2026-01-26

Reject